# Understanding the Role of Training Regimes in Continual Learning

**Seyed Iman Mirzadeh**
Washington State University, USA
`seyediman.mirzadeh@wsu.edu`

**Mehrdad Farajtabar**
DeepMind, USA
`farajtabar@google.com`

**Razvan Pascanu**
DeepMind, UK
`razp@google.com`

**Hassan Ghasemzadeh**
Washington State University, USA
`hassan.ghasemzadeh@wsu.edu`

## Abstract

*Catastrophic forgetting* affects the training of neural networks, limiting their ability to learn multiple tasks sequentially. From the perspective of the well established plasticity-stability dilemma, neural networks tend to be overly plastic, lacking the stability necessary to prevent the forgetting of previous knowledge, which means that as learning progresses, networks tend to forget previously seen tasks. This phenomenon coined in the *continual learning* literature, has attracted much attention lately, and several families of approaches have been proposed with different degrees of success. However, there has been limited prior work extensively analyzing the impact that different training regimes – learning rate, batch size, regularization method– can have on forgetting. In this work, we depart from the typical approach of altering the learning algorithm to improve stability. Instead, we hypothesize that the geometrical properties of the local minima found for each task play an important role in the overall degree of forgetting. In particular, we study the effect of dropout, learning rate decay, and batch size, on forming training regimes that *widen* the tasks' local minima and consequently, on helping it not to forget catastrophically. Our study provides practical insights to improve stability via simple yet effective techniques that outperform alternative baselines.[1]

## 1 Introduction

We study the *continual learning* problem, where a neural network model should learn a sequence of tasks rather than a single one. A significant challenge in continual learning (CL) is that during training on each task, the data from previous ones are unavailable. One consequence of applying typical learning algorithms under such a scenario is that as the model learns newer tasks, the performance of the model on older ones degrades. This phenomenon is known as *"catastrophic forgetting"* [52].

This forgetting problem is closely related to the *"stability-plasticity dilemma"* [53], which is a common challenge for both biological and artificial neural networks. Ideally, a model needs plasticity to obtain new knowledge and adapt to new environments, while it also requires stability to prevent forgetting the knowledge from previous environments. If the model is very plastic but not stable, it can learn fast, but it also forgets quickly. Without further modifications in training, a naively trained neural network tends to be plastic but not stable. Note that plasticity in this scenario does not necessarily imply that neural nets can learn new tasks *efficiently*. In fact, they tend to be extremely data inefficient. By being plastic, we mean a single update can change the function considerably.

With the recent advances in the deep learning field, continual learning has gained more attention since the catastrophic forgetting problem poses a critical challenge for various applications [43, 37]. A growing body of research has attempted to tackle this problem in recent years [58, 72, 57, 29].

Despite the tangible improvements in the continual learning field, the core problem of catastrophic forgetting is still under-studied. In particular, a variety of neural network models and training approaches have been proposed, however, to the best of our knowledge, there has been little work on systematically understanding the effect of common training regimes created by varying dropout regularization, batch size, and learning rate on overcoming catastrophic forgetting[2]. Fig. 1 shows how significantly these techniques can overcome catastrophic forgetting.

In this work, we explore the catastrophic forgetting problem from an optimization and loss landscape perspective (Section 3) and hypothesize that the geometry of the local minima found for the different learned tasks correlates with the ability of the model to not catastrophically forget. Empirically we show how a few well-known techniques, such as dropout and large learning rate with decay and shrinking batch size, can create a training regime to affect the stability of neural networks (Section 4). Some of them, like dropout, had been previously proposed to help continual learning [22, 54]. However, in this work, we provide an alternative justification of why these techniques are effective. Crucially, we empirically show that jointly with a carefully tuned learning rate schedule and batch size, these simple techniques can outperform considerably more complex algorithms meant to deal with continual learning (Section 5). Our analysis can be applied to any other training technique that widens the tasks' local minima or shrinks the distance between them.

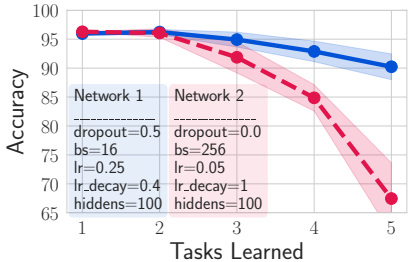

Figure 1: For the same architecture and dataset (Rotation MNIST) and only changing the training regime, the forgetting is reduced significantly at the cost of a relatively small accuracy drop on the current task. Refer to appendix C for details.

Our work shows that plain neural networks can be much stronger baselines for continual learning than previously thought, provided that we use the right hyperparameters. Moreover, the choice for the hyperparameters is orthogonal to other continual learning methods and can be integrated with these methods, as we show in Appendix C.8.

## 2 Related work

Several continual learning methods have been proposed to tackle catastrophic forgetting. Following [43], we categorize these algorithms into three general groups.

The first group consists of replay based methods that build and store a memory of the knowledge learned from old tasks [48, 62, 82, 68, 63], known as *experience replay*. iCaRL [61] learns in a class-incremental way by having a fixed memory that stores samples that are close to the center of each class. Averaged Gradient Episodic Memory (A-GEM) [7] is another example of these methods which build a dynamic episodic memory of parameter gradients during the learning process while ER-Reservoir [9] uses a Reservoir sampling method as its selection strategy.

The methods in the second group use explicit regularization techniques to supervise the learning algorithm such that the network parameters are consistent during the learning process [41, 80, 44, 1, 42]. As a notable work, Elastic weight consolidation (EWC) [41], uses the Fisher information matrix as a proxy for weights' importance and guides the gradient updates. They are usually inspired by a Bayesian perspective [56, 71, 67, 13, 64]. With a frequentist view, some other regularization based methods have utilized gradient information to protect previous knowledge [14, 24, 79]. For example, Orthogonal Gradient Descent (OGD) [14] uses the projection of the prediction gradients from new tasks on the subspace of previous tasks' gradients to maintain the learned knowledge.

Finally, in parameter isolation methods, in addition to potentially a shared part, different subsets of the model parameters are dedicated to each task [65, 78, 35, 60, 46]. This approach can be viewed

as a flexible gating mechanism, which enhances stability and controls the plasticity by activating different gates for each task. [50] proposes a neuroscience-inspired method for a context-dependent gating signal, such that only sparse, mostly non-overlapping patterns of units are active for any one task. PackNet [49] implements a controlled version of gating by using network pruning techniques to free up parameters after finishing each task.

While continual learning is broader than just solving the catastrophic forgetting, in this work we will squarely focus on catastrophic forgetting, which has been an important aspect, if not the most important one, of research for Continual Learning in the last few years. Continual Learning as an emerging field in Artificial Intelligence is connected to many other areas such as Meta Learning [4, 35, 24, 62], Few Shot Learning [75, 20], Multi-task and Transfer Learning [24, 35] and the closely related problem of exploring task boundary or task detection [60, 2, 25, 36]. Very recently, Mirzadeh et al. [55] have studied the mode connectivity of continual learning and multitask learning minima. Moreover, Wallingford et al. [73] proposed a framework for integration of solutions across supervised learning, few-shot learning, continual learning, and efficient machine learning to facilitate the research in the intersection of these fields.

## 3   Forgetting during training

Let us begin this section by introducing some notation to express the effect of *forgetting* during the sequential learning of tasks. For simplicity, let us consider the supervised learning case, which will be the focus of this work. We consider a sequence of $K$ tasks $\mathcal{T}_k$ for $k \in \{1, 2, \ldots, K\}$. Let $\mathcal{W} \in \mathbb{R}^d$ be the parameter space for our model. The total loss on the training set for task $k$ is denoted by

$$L_k(w) = \mathbb{E}[\ell_k(w; x, y)] \approx \frac{1}{|\mathcal{T}_k|} \sum_{(x,y) \in \mathcal{T}_k} , \ell_k(w; x, y) \tag{1}$$

where, the expectation is over the data distribution of task $k$ and $\ell_k$ is a differentiable non-negative loss function associated with data point $(x, y)$ for task $k$. In the continual learning setting, the model learns sequentially, without access to examples of previously seen tasks. For simplicity and brevity, let us focus on measuring the *forgetting* in continual learning with two tasks. It is easy to extend these findings to more tasks.

Let $w_1^*$ and $w_2^*$ be the convergent or optimum parameters after training has been finished for the first and second task sequentially. We formally define the *forgetting* (of the first task) as:

$$F_1 \triangleq L_1(w_2^*) - L_1(w_1^*). \tag{2}$$

We hypothesize that $F_1$ *strongly correlates with properties of the curvature of $L_1$ around $w_1^*$ and $L_2$ around $w_2^*$.* In what follows, we will formalize this hypothesis.

One important assumption that we rely on throughout this section is that we can use a second-order Taylor expansion of our losses to understand the learning dynamics during the training of the model. While this might seem as a crude approximation in general — for a nonlinear function and non-infinitesimal displacement this approximation can be arbitrarily bad — we argue that the approximation has merit for our setting. In particular, we rely on the wealth of observations for overparametrized models where the loss tends to be very well behaved and almost convex in a reasonable neighborhood around their local minima. E.g. for deep linear models this property has been studied in [66]. Works as [11, 23] make similar claims for generic models. [31] also corroborates that within the NTK regime learning is well behaved. In continual learning, similar strong assumptions are made by most approaches that rely on approximating the posterior on the weights by a Gaussian [41] or a first-order approximation of the loss surface around the optimum [14]. Armed with this analytical tool, to compute the forgetting, we can approximate $L_1(w_2^*)$ around $w_1^*$:

$$L_1(w_2^*) \approx L_1(w_1^*) + (w_2^* - w_1^*)^\top \nabla L_1(w_1^*) + \frac{1}{2}(w_2^* - w_1^*)^\top \nabla^2 L_1(w_1^*)(w_2^* - w_1^*) \tag{3}$$

$$\approx L_1(w_1^*) + \frac{1}{2}(w_2^* - w_1^*)^\top \nabla^2 L_1(w_1^*)(w_2^* - w_1^*), \tag{4}$$

where, $\nabla^2 L_1(w_1^*)$ is the Hessian for loss $L_1$ at $w_1^*$ and the last equality holds because the model is assumed to converge to a stationary point where gradient's norm vanishes, thus $\nabla L_1(w_1^*) \approx \mathbf{0}$.

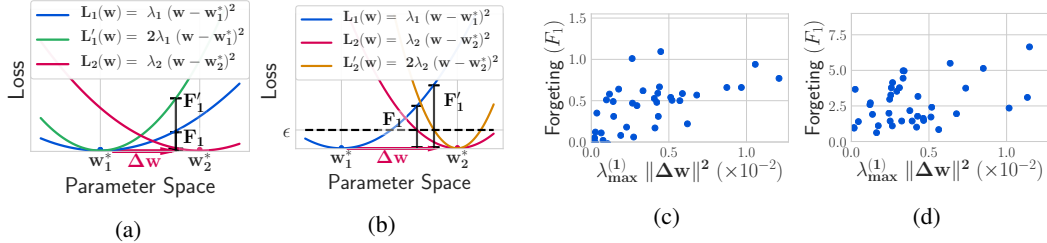

Figure 2: **(a)**: For a fixed $\Delta w$, the wider the curvature of the first task, the less the forgetting. **(b)**: The wider the curvature of the second task, the smaller $\|\Delta w\|$. **(c) and (d):** Empirical verification of (5) for Rotated MNIST and Permuted MNIST, respectively.

Under the assumption that the critical point is a minimum (or the plateau we get stuck in surrounds a minimum), we know that the Hessian needs to be positive semi-definite. Defining $\Delta w = w_2^* - w_1^*$ as the relocation vector, we can bound the forgetting $F_1$ as follows:

$$F_1 = L_1(w_2^*) - L_1(w_1^*) \approx \frac{1}{2} \Delta w^\top \nabla^2 L_1(w_1^*) \Delta w \leq \frac{1}{2} \lambda_1^{max} \|\Delta w\|^2 , \qquad (5)$$

where $\lambda_1^{max}$ is the maximum eigenvalue of $\nabla^2 L_1(w_1^*)$. Fig. 2a shows how wider $L_1$ (lower $\lambda_1^{max}$) leads to less forgetting, both in terms of an illustrative example as well as showing empirical evidence for this relationship on Rotated MNIST and Permuted MNIST.

A few notes on this bound. First, the bound is achieved when $\Delta w$ is co-aligned to the eigenvector corresponding to $\lambda_1$. Given that the displacement is generated by the learning process on task 2, we can think of it as a random vector with respect to the eigenvectors of the Hessian of the first task. We know, therefore, that the tightness of the bound correlates with the dimensionality of $w$. In the extreme one-dimensional case, the Hessian matrix becomes a scalar given by $\lambda_1^{max}$, and the bound is exact. As we increase the number of dimensions, the probability of two vectors to be perpendicular goes to 1. Hence in high-dimensional spaces is more likely for the bound to be relatively loose and for the entire spectrum of eigenvalues to play a much more important role in defining $F_1$. Namely, as the number of eigenvalues with low magnitude increases, the more likely it is for $F_1$ to be small. Assuming that the displacement vector is equally distributed over all the eigenvectors, then the trace of the Hessian will correlate stronger with $F_1$ than the largest eigenvalue. However, reasoning in terms of the spectrum can be impractical (note, for example, that one can not trivially re-write the bound in terms of the trace). So we believe it is useful to think about $\lambda_1^{max}$ as long as any conclusion is contextualized correctly and the training regime we consider, implicitly, is aimed at lowering the entire spectrum, not just the largest eigenvalue.

We also want to highlight $\lambda_1^{max}$ has been used previously to describe the *width of a local minima* [28, 39], with similar notes regarding the role of the entire spectrum [12, 39]. This property is central to the wide/narrow minima hypothesis for why neural networks generalize well. Our hypothesis is not tied to the generalization of wide minima, but we rely on the same or at least a very related concept of *width*. Therefore, to reduce forgetting, each task should push its learning towards *wider minima* and can employ the same techniques used to widen the minima to improve generalization.

Resuming from Eq. 5, controlling the Hessian spectrum without controlling the norm of the displacement, however, might not ensure that $F_1$ is minimized.

$\|\Delta w\|$ is technically controlled by the subsequent tasks. We first notice, empirically, that enforcing widening the minima of the next task (for the same reason of reducing forgetting on itself) inhibits additionally forgetting for the first task (see Table 1; not stable/plastic means relying on training regimes that encourage/do not encourage wide minima. We empirically estimate the width of minima as well, see appendix C for details). We make the observation that the width of the minima (norm of the eigenvalues) correlates with the norm of the weights. Hence the solutions in the stable learning regime tend to be closer to **0**, which automatically decrease $\|\Delta w\|$.

Table 1: Disentangling the forgetting on Permuted MNIST. Details are left to appendix C.

| Task 1 | Task 2 | Forgetting (F1) |
|---|---|---|
| Stable | Stable | $1.61 \pm (0.48)$ |
| Stable | Plastic | $4.77 \pm (1.72)$ |
| Plastic | Stable | $12.45 \pm (1.58)$ |
| Plastic | Plastic | $19.37 \pm (1.79)$ ) |

Additionally, $\|\Delta w\|$ relates to $\lambda_2^{max}$ also due to typical learning terminating near a minima, rather than at the minima. Refer to fig. 2b for an illustration. Specifically, the convergence criterion is usually satisfied in the $\epsilon$-neighborhood of $w_2^*$. If we write the second order Taylor approximation of $L_2$ around $w_2^*$, we get:

$$L_2(\hat{w}_2) - L_2(w_2^*) \approx \frac{1}{2}(\hat{w}_2 - w_2^*)^\top \nabla^2 L_2(\hat{w}_2)(\hat{w}_2 - w_2^*) \leq \frac{1}{2}\lambda_2^{max} \|w_2^* - \hat{w}_2\|^2 \leq \epsilon, \quad (6)$$

where, the first equality holds since $\nabla L_2(w_2^*) = 0$. Thus, by decreasing $\lambda_2^{max}$, $\hat{w}_2$ can be reached further from $w_2^*$ since the $\epsilon$-neighborhood is larger, and closer to $w_1^*$. A more formal analysis is given in the appendix. Note that as we enforce the error on task 2 to be lower, the argument above weakens. In the limit, if we assume you converged on task 2, the distance does not depend on curvature, just $\|w_1^* - w_2^*\|$. However, the choice of which minima $w_2^*$ learning prefers will still affect the distance, and as argued above, if wider minima tend to be closer to 0, then they tend to be closer to each other too. Collating all of these observations together we propose the following hypothesis:

**Hypothesis**. *The amount of forgetting that a neural network exhibits from learning the tasks sequentially, correlates with the geometrical properties of the convergent points. In particular, the wider these minima are, the less forgetting happens.*

We empirically verify the relationship between forgetting and the upper bound derived in E.q. (5). We approximate the Hessian with the largest eigenvalue of the loss function. The results in two common continual learning benchmarks is shown in Figures 2c and 2d. In the figure, the dots represent different neural network training regimes with different settings (e.g., with and without dropout, with and without learning rate decay, different initial learning rates, different batch sizes, different random initialization). See section 4 to find out how these techniques can lead to different loss geometries. All of the models have roughly 90% accuracy on task 2. We can see that our derived measure has high correlation with the forgetting.

## 4 Training Regimes: techniques affecting stability and forgetting

In this section, we describe a set of widely used techniques that are known to affect the width of the minima (eigenspectrum of Hessian) as well as the length of the path taken by learning ($\|\Delta w\|$). These observations had been generally made with respect to improving generalization, following the wide/narrow minima hypothesis. Based on the argumentation of the previous section, we believe these techniques can have an important role in affecting forgetting as well. In the following section, we will validate this through solid empirical evidence that agrees with our stated hypothesis.

### 4.1 Optimization setting: learning rate, batch size, and optimizer

There has been a large body of prior work studying the relationship between the learning rate, batch size, and generalization. One common technique of analysis in this direction is to measure the largest eigenvalues of the Hessian of the loss function, which quantifies the local curvature around minima [39, 16, 33, 32, 33]. Followed by the work by Keskar et al. [39], several other papers studied the correlation between minima wideness and generalization [32, 51, 33].

The learning rate and batch size influence both the endpoint curvature and the whole trajectory [77, 17, 45]. A high learning rate or a small batch size limits the maximum spectral norm along the path found by SGD from the beginning of training [33]. This is further studied by Jastrzebski et al. [34], showing that using a large learning rate in the initial phase of training reduces the variance of the gradient, and improves the conditioning of the covariance of gradients which itself is close to the Hessian in terms of the largest eigenvalues [81].

Although having a higher learning rate tends to be helpful since it increases the probability of converging to a wider minima [34], considering a continual optimization problem, we can see it has another consequence: it contributes to the rate of change (i.e., $\Delta w$ in (5)). Using a higher learning rate means applying a higher update to the neural network weights. Therefore, since the objective function changes thorough time, having a high learning rate is a double-edged sword. Intuitively speaking, decreasing the learning rate across tasks prevents the parameters from going far from the current optimum, which helps reduce forgetting. One natural solution could be to start with a high initial learning rate for the first task to obtain a wide and stable minima. Then, for each subsequent task, slightly decrease the learning rate but also decrease the batch-size instead, as suggested in [69].

Regarding the choice of an optimizer, we argue for the effectiveness of SGD in continual learning setting compared to adaptive optimizers. Although the adaptive gradient methods such as Adam [40] tend to perform well in the initial phase of training, they are outperformed by SGD at later stages of training due to generalization issues [38, 10]. Wilson et al. [76] show that even for a toy quadratic problem, Adam generalizes provably worse than SGD. Moreover, Ge et al. [19] study the effectiveness of exponentially decaying learning rate and show that in its final iterate, it can achieve a near-optimal error in stochastic optimization problems.

**Connection to continual learning.** The effect of learning rate and batch-size has not been directly studied in the context of continual learning, to the extent of our knowledge. However, we find that the reported hyper-parameters in several works match our findings. Using a small batch size is very common across the continual learning methods. OGD [14] uses a small batch size of 10, similar to several other works [9, 7, 8]. EWC [41] uses a batch size of 32 and also the learning rate decay of $0.95$. iCaRL [61] starts from a large learning rate and decays the learning at certain epochs exponentially by a factor of $0.2$, while it uses a larger batch size of $128$. Finally, PackNet [49] also reports using a learning rate decay by a factor of $0.1$. When it comes to choosing the optimizer, the literature mostly prefers SGD with momentum over the adaptive gradient methods. With the exception of VCL [56], which uses Adam, several other algorithms such as A-GEM, OGD, EWC, iCaRL, and PackNet use the standard SGD.

## 4.2 Regularization: dropout and weight decay

We relate the theoretical insights on dropout and $L_2$ regularization (weight decay) to our analysis in the previous section. We first argue for the effectiveness of dropout, and then we discuss why $L_2$ regularization might hurt the performance in a continual learning setting.

Dropout [27] is a well-established technique in deep learning, which is also well-studied theoretically [3, 70, 26, 18, 74]. Wei et al. [74] showed that dropout has both implicit and explicit but entangled regularization effects: More specifically, they provided an approximation for the explicit effect on the $i$-th hidden layer (denoted by $h_i$) under input $x$ by: $(p/p-1)[\nabla^2_{h_i} L]^T [\text{diag}(h_i^2)]$, where $p$ is the dropout probability, $L$ is the loss function, and $\text{diag}(v)$ is a diagonal matrix of vector $v$. This term encourages the flatness of the minima since it minimizes the second derivative of the loss with respect to the hidden activation (that is tightly correlated with the curvature with respect to weights). Thus, dropout regularization reduces the R.H.S of Eq. (5). Note that by regularizing the activations norm, it also pushes down the norm of the weights, hence encouraging to find a minima close to 0, which in turn could reduce the norm of $\Delta w$. Intuitively, we can understand this effect also from the tendency of dropout to create redundant features. This will reduce the effective number of dimensions of the representation, increasing the number of small-magnitude eigenvalues for the Hessian. As a consequence, gradient updates of the new tasks are less likely to lie on the space spanned by significant eigendirections of previous losses, which results in lesser forgetting.

With respect to $L_2$ regularization, while intuitively it should help, we make two observations. First, dropout is data-dependent, while $L_2$ is not. That means balancing the effect of regularization with learning is harder, and in practice, it seems to work worse (both for the currently learned task and for reducing forgetting while maintaining good performance on the new task). Secondly, when combined with Batch Normalization [30], $L_2$ regularization leads to an Exponential Learning Rate Schedule [47], meaning that in each iteration, multiplying the initial learning rate by $(1 + \alpha)$ for some $\alpha > 0$ that depends on the momentum and weight decay rate. Hence, using $L_2$ regularization is equivalent to increasing the learning rate overall, potentially leading to larger displacements $\Delta w$.

**Connection to continual learning.** To the best of our knowledge, the work by Goodfellow et al. [22] is the first to empirically study the importance of the dropout technique in the continual learning setting. They hypothesize that dropout increases the optimal size of the network by regularizing and constraining the capacity to be just barely sufficient to perform the first task. However, by observing some inconsistent results on dissimilar tasks, they suggested dropout may have other beneficial effects too. Very recently, [54] studied the effectiveness of dropout for continual learning from the gating perspective. Our work extends their analysis in a more general setting by studying the regularization effect of dropout and its connection to loss landscape. Finally, [43] conducted a comprehensive empirical study on the effect of weight decay and dropout on the continual learning performance and reported that the model consistently benefits from dropout regularization as opposed to weight decay which results in increased forgetting and lower performance on the final model.

# 5   Experiments and results

In this section, after explaining our experimental setup, we show the relationship between the curvature of the loss function and the amount of forgetting. We use the terms *Stable* and *Plastic (Naive)* to distinguish two different training regimes. The stable network (or stable SGD) exploits the dropout regularization, large initial learning rate with exponential decay schedule at the end of each task, and small batch size, as explained in Sec. 4. In contrast, the plastic (naive) SGD model does not exploit these techniques. In the second experiment, we challenge the stable network and compare it to various state of the art methods on a large number of tasks and more difficult benchmarks. In Appendix C.8, we will show that the stable training regime can be integrated into other continual learning methods and improve their performance significantly.

## 5.1   Experimental setup

Here, we discuss our experimental methodologies. The decisions regarding the datasets, network architectures, continual learning setup (e.g., number of tasks, training epochs per task), hyper-parameters, and evaluation metrics are chosen to be consistent with several other studies [7, 9, 8, 14], making it easy to compare our results. For all experiments, we report the average and standard deviation over five runs, each with a different random seed. For brevity, we include the detailed hyper-parameters, the code, and instructions for reproducing the results in the supplementary file.

**Datasets.** We perform our experiments on three standard continual learning benchmarks: Permuted MNIST [22], Rotated MNIST, and Split CIFAR-100. While we agree with [15] regarding the drawbacks of Permuted MNIST in continual learning settings, we believe for consistency with other studies, it is essential to report the results on this dataset as well. Moreover, we report our results on Rotated MNIST and CIFAR-100 that are more challenging and realistic datasets for continual learning benchmarks, once the number of tasks is large. Each task of permuted MNIST is generated by random shuffling of the pixels of images such that the permutation is the same for the images of the same task, but different across different tasks. Rotated MNIST is generated by the continual rotation of the MNIST images where each task applies a fixed random image rotation (between 0 and 180 degrees) to the original dataset. Split CIFAR-100 is a variant of the CIFAR-100 where each task contains the data from 5 random classes (without replacement) out of the total 100 classes.

**Models.** In our first experiment (Sec. 5.2), we evaluate the continual learning performance over five sequential tasks to provide fine-grained metrics for each task. For this experiment, we use a feed-forward neural network with two hidden layers, each with 100 ReLU neurons and use the deflated power iteration for computing eigenvalues [21]. For the second experiment (Sec. 5.3), we scale the experiments to 20 tasks and use a two-layer network with 256 ReLU neurons in each layer for MNIST datasets, and a ResNet18, with three times fewer feature maps across all layers for CIFAR experiments. These architectures have been previously chosen in several studies [7, 9, 14, 8].

**Evaluation.** We use two metrics from [6, 7, 9] to evaluate continual learning algorithms when the number of tasks is large.
**(1) Average Accuracy**: The average validation accuracy after the model has been trained sequentially up to task $t$, defined by:

$$\mathbf{A_t} = \frac{1}{t} \sum_{i=1}^{t} a_{t,i} \tag{7}$$

where, $a_{t,i}$ is the validation accuracy on dataset $i$ when the model finished learning task $t$.
**(2) Average Forgetting**: The average forgetting after the model has been trained sequentially on all tasks. Forgetting is defined as the decrease in performance at each of the tasks between their peak accuracy and their accuracy after the continual learning experience has finished. For a continual learning dataset with $T$ sequential tasks, it is defined by:

$$\mathbf{F} = \frac{1}{T-1} \sum_{i=1}^{T-1} \max_{t \in \{1, \dots, T-1\}} \left( a_{t,i} - a_{T,i} \right) \tag{8}$$

To prevent confusion, we note that this definition of forgetting is different from what we studied in Section 3. Here, the average forgetting is an evaluation metric that is computed from validation accuracy of the model.

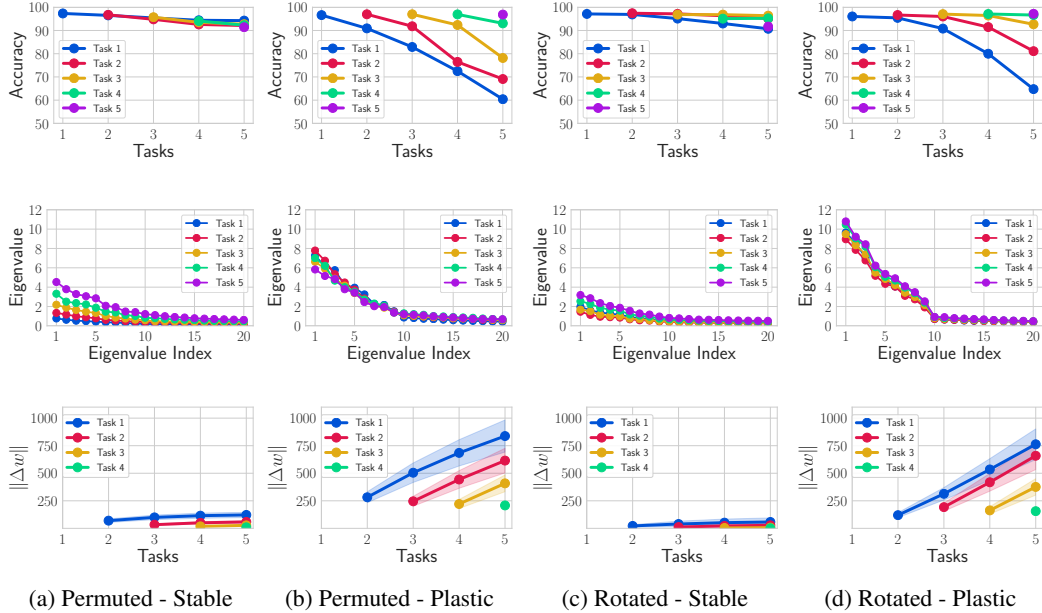

(a) Permuted - Stable    (b) Permuted - Plastic    (c) Rotated - Stable    (d) Rotated - Plastic

Figure 3: Comparison of training regimes for MNIST datasets: **(Top)** *Evolution of validation accuracy for each task*: stable networks suffer less from catastrophic forgetting. **(Middle)** *The spectrum of the Hessian for each task*: the eigenvalues in stable networks are significantly smaller. **(Bottom)** *The degree of parameter change*: in stable networks, the parameters remain closer to the their initial values after learning each task.

## 5.2  Stable versus Plastic networks

Here, we verify the significance of the training regime in continual learning performance (Sec 3.) and demonstrate the effectiveness of the stability techniques (Sec 4.) in reducing the forgetting.

Each row of Figure 3, represents one of three related concepts for each training regime on each dataset. First, the top row shows the evolution of accuracy on validation sets of each task during the continual learning experience. For instance, the blue lines in this row show the validation accuracy of task 1 throughout the learning experience. In the Middle row, we show the twenty sharpest eigenvalues of the curvatures of each task. In the bottom row, we measure the $\ell_2$ distance of network parameters between the parameters learned for each task, and the parameters learned for subsequent tasks.

Aligned with our analysis in Section 3, we show that in contrast to the *plastic* regime, the *stable* training reduces the catastrophic forgetting (Fig. 3 (Top)) thanks to (1) decreasing the *curvature* (Fig. 3 (Middle)) and (2) shrinking the change of parameters (Fig. 3 (Bottom)).

## 5.3  Comparison with other methods

In this experiment, we show that the stable network is a strong competitor for various continual learning algorithms. In this scaled experiment, we increase the number of tasks from 5 to 20, and provide results for Split CIFAR-100, which is a challenging benchmark for continual learning algorithms. The episodic memory size for A-GEM and ER-Reservoir is limited to be one example per class per task (i.e., 200 examples for MNIST experiments and 100 for CIFAR-100), similar to [9, 8]. To have a consistent naming with other studies, in this section, we use the word "Naive" to describe a plastic network in our paper. To evaluate each algorithm, we measure the average accuracy and forgetting (i.e., $A_t$ and $F$ in Sec. 5.1).

Table 2 compares these metrics for each method once the continual learning experience is finished (i.e., after learning task 20). Moreover, Fig. 4 provides a more detailed picture of the average accuracy during the continual learning experience. To show that stable networks suffer less from catastrophic forgetting, we provide a comparison of the first task's accuracy in the appendix.

Table 2: Comparison of the average accuracy and forgetting of several methods on three datasets.

| Method | Memoryless | Permuted MNIST | | Rotated MNIST | | Split CIFAR100 | |
|---|---|---|---|---|---|---|---|
| | | Accuracy | Forgetting | Accuracy | Forgetting | Accuracy | Forgetting |
| Naive SGD | ✓ | 44.4 (±2.46) | 0.53 (±0.03) | 46.3 (±1.37) | 0.52 (±0.01) | 40.4 (±2.83) | 0.31 (±0.02) |
| EWC | ✓ | 70.7 (±1.74) | 0.23 (±0.01) | 48.5 (±1.24) | 0.48 (±0.01) | 42.7 (±1.89) | 0.28 (±0.03) |
| A-GEM | ✗ | 65.7 (±0.51) | 0.29 (±0.01) | 55.3 (±1.47) | 0.42 (±0.01) | 50.7 (±2.32) | 0.19 (±0.04) |
| ER-Reservoir | ✗ | 72.4 (±0.42) | 0.16 (±0.01) | 69.2 (±1.10) | 0.21 (±0.01) | 46.9 (±0.76) | 0.21 (±0.03) |
| Stable SGD | ✓ | **80.1 (±0.51)** | **0.09 (±0.01)** | **70.8 (±0.78)** | **0.10 (±0.02)** | **59.9 (±1.81)** | **0.08 (±0.01)** |
| Multi-Task Learning | N/A | 86.5 (±0.21) | 0.0 | 87.3(±0.47) | 0.0 | 64.8(±0.72) | 0.0 |

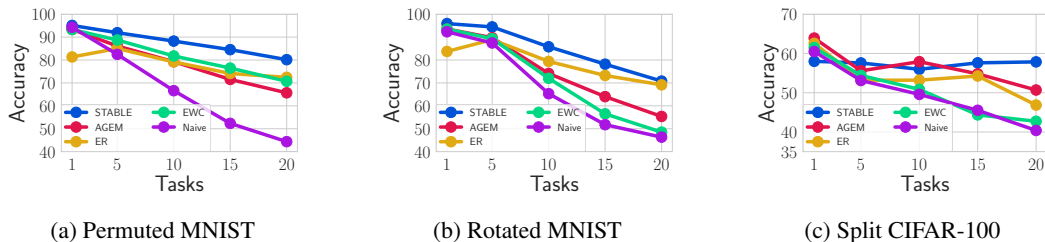

(a) Permuted MNIST        (b) Rotated MNIST        (c) Split CIFAR-100

Figure 4: Evolution of the average accuracy during the continual learning experience with 20 tasks

While our stable network performs consistently better than other algorithms, we note that our proposed techniques are orthogonal to other works and can also be incorporated in them, as we show in Appendix C.8

# 6 Conclusion

In this work, we have revisited the catastrophic forgetting problem from loss landscapes and optimization perspective and identify learning regimes and training techniques that contribute to the forgetting. The analytical insights yielded a series of effective and practical techniques that can reduce forgetting and increase the stability of neural networks in maintaining previous knowledge.

We have studied these techniques through the lens of optimization by studied the wideness of the loss surfaces around the local minima. However, they might have other confounding factors for reducing catastrophic forgetting as well. We call for more theoretical research to further their role in demystifying trading off the stability plasticity dilemma and its effect on continual learning.

Finally, we have empirically observed that these simple techniques proved to be more effective than some of the recent approaches (e.g., regularization based methods, or memory-based methods) but are orthogonal to them in the sense that our practical recommendations and provided insights on loss perspective can be incorporated to them.

## Broader Impact

Continual Learning aims for effectively training a model from sequential tasks while making sure the model maintains a reasonable performance on the previous ones. It's an integral part of Artificial General Intelligence (AGI) that reduces the cost of retraining (time, computation, resources, energy) and mitigates the need for storing all previous data to respect users' privacy concerns better. Reducing catastrophic forgetting may potentially risk privacy for data that are explicitly wanted to be forgotten. This calls for more future research into formalizing and proposing continual learning agents that allow the identifiable parts of data to be forgotten, but the general knowledge is maintained. The research presented in this paper can be used for many different application areas and a particular use may have both positive or negative implications. Besides those, we are not aware of any immediate short term negative impact.

## Acknowledgment

SIM and HG acknowledge support from the United States National Science Foundation through grant CNS-1750679. The authors thank Anonymous Reviewers, Jonathan Schwarz, Sepehr Sameni, Hooman Shahrokhi, and Mohammad Sadegh Jazayeri for their valuable comments and feedback.

## Footnotes

[1] The code is available at: https://github.com/imirzadeh/stable-continual-learning

[2]Potential exceptions being the early work of Goodefellow et. al [22] and a recent one by Mirzadeh et. al [54]

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
