[Supplementary Material]

# Supplementary material

In this document, we present the materials that were excluded or summarized due to space limitation in the main text. It is organized as follows:

**Appendix A** extends our analysis in Section 3 of the main text regarding the forgetting in continual learning.

**Appendix B** provides further information for our experimental setup and the hyper-parameters used for each experiment. Note that, in addition to this document, we provide our code with scripts to reproduce the results for each experiment.

**Appendix C** includes additional experiments and results. More specifically, it includes:

- Additional information about Figure 1 such as accuracy on all tasks.
- Extended version of Table 1 with detailed discussion.
- Additional results regarding the norm of parameters in our first experiment (Section. 5.2)
- Additional results regarding the comparison of the first task accuracy for algorithms in our second experiment (Section. 5.3).
- A **new experiment** in Appendix C.8, where we apply the stability techniques for other methods such as A-GEM and EWC. We demonstrate that these methods can enjoy a performance boost if they use a stable training regime.

## A    Further analysis on forgetting

In this section, we extend our analysis in Section 3 of the main paper.

Let $w_1^*$ and $w_2^*$ still be the optimal points for the tasks while $\hat{w}_1$ and $\hat{w}_2$ are the convergent or (near-) optimum parameters after training has been finished for the first and second task, sequentially. We usually use Stochastic Gradient Descent (SGD) or its many variants to find a low-error plateau for the optimization procedure. We can define the *forgetting* (of the first task) using the convergent point too:

$$F_1 \triangleq L_1(\hat{w}_2) - L_1(\hat{w}_1). \tag{9}$$

We can approximate $L_1(\hat{w}_2)$ and $L_1(\hat{w}_1)$ around $w_1^*$:

$$L_1(\hat{w}_1) \approx L_1(w_1^*) + (\hat{w}_1 - w_1^*)^\top \nabla L_1(w_1^*) + \frac{1}{2}(\hat{w}_1 - w_1^*)^\top \nabla^2 L_1(w_1^*)(\hat{w}_1 - w_1^*) \tag{10}$$

$$\approx L_1(w_1^*) + \frac{1}{2}(\hat{w}_1 - w_1^*)^\top \nabla^2 L_1(w_1^*)(\hat{w}_1 - w_1^*), \tag{11}$$

where, $\nabla^2 L_1(w_1^*)$ is the Hessian for loss $L_1$ at $w_1^*$ and the last equality holds because the model is assumed to converge to critical point or it had stopped in a plateau where gradient's norm vanishes, thus $\nabla L_1(w_1^*) \approx \mathbf{0}$. Similarly,

$$L_1(\hat{w}_2) \approx L_1(w_1^*) + \frac{1}{2}(\hat{w}_2 - w_1^*)^\top \nabla^2 L_1(w_1^*)(\hat{w}_2 - w_1^*). \tag{12}$$

Defining $\Delta w = \hat{w}_2 - \hat{w}_1$ as the relocation vector we compute the forgetting according to Eq. (9):

$$F_1 \approx \frac{1}{2}(\hat{w}_2 - w_1^*)^\top \nabla^2 L_1(w_1^*)(\hat{w}_2 - w_1^*) - \frac{1}{2}(\hat{w}_1 - w_1^*)^\top \nabla^2 L_1(w_1^*)(\hat{w}_1 - w_1^*) \tag{13}$$

$$= \frac{1}{2}((\hat{w}_2 - w_1^*) - (\hat{w}_1 - w_1^*)))\nabla^2 L_1(w_1^*)((\hat{w}_2 - w_1^*) + (\hat{w}_1 - w_1^*))) \tag{14}$$

$$\approx \frac{1}{2}\Delta w^\top \nabla^2 L_1(w_1^*)\Delta w, \tag{15}$$

where in the last equality we used $\|\hat{w}_1 - w_1^*\| \ll \|\hat{w}_2 - w_1^*\|$. Under the assumption that the critical point is a minima (or the plateau we get stuck surrounding a minima), we know that the Hessian needs to be positive semi-definite. We can use this property further to bound $F_1$ as follows:

$$F_1 = L_1(\hat{w}_2) - L_1(\hat{w}_1) \approx \frac{1}{2}\Delta w^\top \nabla^2 L_1(w_1^*)\Delta w \leq \frac{1}{2}\lambda_1^{max} \|\Delta w\|^2, \tag{16}$$

where $\lambda_1^{max}$ is the maximum eigenvalue of $\nabla^2 L_1(w_1^*)$. Fig. 5a shows how wider $L_1$ (lower $\lambda_1^{max}$) leads to less forgetting.

Controlling the spectrum of the Hessian without controlling the norm of the displacement can not ensure that $F_1$ is minimized.

Assume the algorithm has already converged to a plateau where gradient's norm vanishes for $L_1$ and stopped at $\hat{w}_1$ and is about to optimize $L_2$. The weights are updated according to $\eta \nabla L_2(w)$ iteratively until a fixed number of iterations, or a minimum (validation) loss, or minimum gradient magnitude is achieved. We show that $\|\Delta w\|$ is related to $\lambda_2^{max}$ (Refer to fig. 5b for an illustration). Using the triangle inequality we write:

$$\|\Delta w\| = \|\hat{w}_2 - \hat{w}_1\| \geq \|w_2^* - \hat{w}_1\| - \|\hat{w}_2 - w_2^*\|. \tag{17}$$

Since $\|w_2^* - \hat{w}_1\|$ is constant we only need to bound $\|\hat{w}_2 - w_2^*\|$. We examine two different convergence criterion.

For the case $L_2(\hat{w}_2) - L_2(w_2^*) \leq \epsilon$ is the convergence criterion, we write the second order Taylor approximation of $L_2$ around $w_2^*$:

$$L_2(\hat{w}_2) - L_2(w_2^*) \approx (\hat{w}_2 - w_2^*)^\top \nabla L_2(w_2^*) + \frac{1}{2}(\hat{w}_2 - w_2^*)^\top \nabla^2 L_2(\hat{w}_2)(\hat{w}_2 - w_2^*). \tag{18}$$

Again, $\nabla L_2(w_2^*) = 0$ and the first term in the R.H.S. is dismissed. Moreover the second term in the R.H.S. can be upper bounded by $\frac{1}{2}\lambda_2^{max}\|w_2^* - \hat{w}_2\|^2$. Therefore, one can write the convergence criterion as

$$\frac{1}{2}\lambda_2^{max}\|w_2^* - \hat{w}_2\|^2 \leq \epsilon \implies \|w_2^* - \hat{w}_2\| \leq \frac{2\sqrt{\epsilon}}{\lambda_2^{max}}. \tag{19}$$

Combining the above with (17) we get:

$$\|\Delta w\| \geq C - \frac{2\sqrt{\epsilon}}{\lambda_2^{max}}, \tag{20}$$

where, $C = \|w_2^* - \hat{w}_1\|$ is constant. Therefore, by decreasing $\lambda_2^{max}$, the lower bound on the $\|\Delta w\|$ decreases and the near-optimum $w_2^*$ can be reached in a closer distance to $w_1^*$. Fig. 5b shows this case.

For the case where $\|\nabla L_2\|(\hat{w}_2) \leq \epsilon$ is the convergence criterion we write the first order Taylor approximation of $\nabla L_2$ around $w_2^*$:

$$\nabla L_2(\hat{w}_2) - \nabla L_2(w_2^*) \approx \nabla^2 L_2(w_2^*)(\hat{w}_2 - w_2^*). \tag{21}$$

Again, $\nabla L_2(w_2^*) = 0$ and the second term in the L.H.S. is dismissed. Moreover, the R.H.S. can be upper bounded by $\lambda_2^{max}\|w_2^* - \hat{w}_2\|$. Therefore, one can write the convergence criterion as

$$\lambda_2^{max}\|w_2^* - \hat{w}_2\| \leq \epsilon \implies \|w_2^* - \hat{w}_2\| \leq \frac{\epsilon}{\lambda_2^{max}}. \tag{22}$$

Combining the above with (17) we get:

$$\|\Delta w\| \geq C - \frac{\epsilon}{\lambda_2^{max}}, \tag{23}$$

where, $C = \|w_2^* - \hat{w}_1\|$ is constant. Therefore, by decreasing $\lambda_2^{max}$, the lower bound on the $\|\Delta w\|$ decreases and the near-optimum $\hat{w}_2$ can be reached within a smaller distance from $\hat{w}_1$.

In practice, we usually fix the number of gradient update $\eta \nabla L_2(w)$ iterations where the learning rate and the gradient of the loss for task 2, play important roles. First, it's already clear that the smaller the learning rate, the higher the chance of finding $\hat{w}_2$ close to $\hat{w}_1$. More importantly, learning decay helps with wider minima by allowing to start with a large learning rate and imposing an exploration phase that increases the chance of finding a wider minima. Therefore, learning rate decay plays a significant role in reducing catastrophic forgetting.

Second, the magnitude of the update is proportional to $\|\nabla L_2(w)\|$ where is lager for higher curvature not only at the minima but in its neighborhood too. Look at Fig. 5b for a simplistic illustration where we assume that the (near-)optimal points for the low and high curvature achieved lie very close to each other. Again, we can intuitively see that the forgetting is correlated with the curvature around the local minimas.

Figure 5: **(a)**: For a fixed $\Delta w$, the wider the curvature of the first task, the less the forgetting. **(b)**: The wider the curvature of the second task, the smaller $\|\Delta w\|$.

# B  Experimental Setup Details

## B.1  Discussion on our experiment design

We would like to note that one fundamental criterion we take into account in our experiments is to facilitate the verification of our results. We achieve this goal by:

1. Reporting the results on common continual learning datasets used in previous studies: For instance, as mentioned in the main text, the majority of continual learning papers to which we compare our results use Permuted MNIST, Rotated MNIST, and Split CIFAR-100 benchmarks. While we acknowledge the fact that continual learning literature can benefit from benchmarks in other domains beyond computer vision, we agree with the current trend that CIFAR-100 with 20 sequential tasks is challenging enough for continual learning.

2. Using similar architectures with other studies: For our first experiment, we used two-layer MLP on MNIST datasets o 5 tasks. This architecture is used in [14, 74]. For the second experiment on 20 tasks, the Resnet architecture described in the text, was used in [7, 9, 8]. The only change we apply to the architecture was to add dropout layers in residual blocks.

3. Providing metrics for accuracy and forgetting: We believe any continual learning work should provide report both mentioned metrics. We also report these metrics, which are also reported in [7, 8, 9].

4. **Releasing the code.** Please refer to our code repository for further instructions for replicating the results. We have implemented stable SGD with PyTorch [59].

## B.2  Continual learning setup

In this section, we review our experimental setup (e.g., number of tasks, number of epochs per task).

In experiment 1 (Section 5.2), our goal was to *elaborate the impact of training regime*. Hence, we found that five tasks and two benchmarks are sufficient for our purpose. Each task in that experiment had 5 training epochs.

In experiment 2 (Section 5.3), we aimed to *demonstrate the performance of stable training regime*, and hence, we chose the setup that is similar to our baselines. We used 20 tasks where each task had 1 training epoch. This setup is used in several studies [7, 9, 8]. We note that we use the task identifiers only in the CIFAR-100 experiment, similar to A-GEM and ER-Reservoir implementations.

In the new experiment in Appendix C.8, the purpose is to *show that the stability techniques can be incorporated into other algorithms* such as A-GEM, EWC, and ER-Reservoir. We use the rotation MNIST with 20 tasks and 1 epoch per task.

Finally, We note that we have used Stochastic Gradient Descent (SGD) with momentum optimizer in all experiments.

### B.3 Hyperparameters in experiments

In this section, we report the hyper-parameters we used in our experiments. For other algorithms (e.g., A-GEM, and EWC), we ensured that our hyper-parameters included the optimal values that the original papers reported. We used grid search for each model to find the best set of parameters detailed below[3]:

**Parameters for experiment 1**

#### B.3.1 Plastic (Naive)

- initial learning rate: [0.25, 0.1, **0.01**, 0.001]
- batch size: 64

#### B.3.2 Stable

- initial learning rate: [0.25, **0.1**, 0.01, 0.001]
- learning rate decay: [0.9, 0.75, **0.4**, 0.25]
- batch size: [**16**, 64]
- dropout: [**0.5**, 0.25]

**Parameters for experiment 2**

#### B.3.3 Naive

- initial learning rate: [0.25, 0.1, **0.01** (MNIST, CIFAR-100), 0.001]
- batch size: [**10**, 64]

#### B.3.4 Stable

- initial learning rate: [0.25, **0.15** (CIFAR-100), **0.10** (MNIST), 0.05, 0.01, 0.005]
- learning rate decay: [0.9, **0.85** (CIFAR-100), 0.7, **0.55** (MNIST), 0.5]
- batch size: [**10**, 64]
- dropout: [**0.25** (MNIST), **0.10** (CIFAR-100), 0.0]

#### B.3.5 EWC

- initial learning rate: [0.25, **0.1** (MNIST, CIFAR-100), 0.01, 0.001]
- batch size: [**10**, 64]
- $\lambda$ (regularization): [1, **10** (MNIST, CIFAR-100), 100]

#### B.3.6 AGEM

- initial learning rate: [0.25, **0.1** (MNIST), **0.01** (CIFAR-100), 0.001]
- batch size: [**10**, 64]

#### B.3.7 ER

- initial learning rate: [0.25, **0.1** (MNIST), **0.01** (CIFAR-100), 0.001]
- batch size: [**10**, 64]

# C  Additional Results

## C.1  Detailed accuracy of the networks in Figure 1

Figure 1 in the introduction (Section 1), compares the first task accuracy between the naive SGD and the stable SGD. Here, we provide the accuracy of all tasks for these two training regimes.

Although in the introduction section we did not introduce the "stable training" term, we note that Network 1 (i.e., the stable network), suffers much less from the catastrophic forgetting of previous tasks compared to Network 2 (i.e., the plastic network), thanks to the training regime, and at the cost of a relatively small drop in the accuracy of the current task.

(a) Network 1 (Stable Network)                          (b) Network 2 (Plastic Network)

Figure 6: Full results for the comparison of accuracy on all tasks for two networks in Figure 1 of the introduction section.

## C.2  Extended version of Table 1 in Section 3: disentangling the stability of different tasks

In this section, we aim to disentangle the role that stable training of the current task 1 plays against stable training of task 2 in decreasing the catastrophic forgetting for task 1.

In table 3, we report the forgetting measure $F_1$ for four different scenarios: training the first task (with/without) stability techniques and training the second task (with/without) these techniques, respectively. The reported "Stable" networks in this table use dropout probability of $0.25$, learning rate decay of $0.4$ and a small batch size of $16$, while "Plastic" networks do not exploit the dropout regularization and learning rate decay, and set the batch size to $256$.

The reported mean and standard deviations are calculated for five different runs with different random seeds. As expected, the case Stable/Stable is the best and Plastic/Plastic is the worst in terms of forgetting. But the interesting observation is the huge difference between Stable/Plastic and Plastic/Stable. This suggests, among other possible explanations, that the wideness of the current task is more important than the wideness of the subsequent tasks for the goal of reducing the forgetting.

Table 3: Disentanglment of forgetting on Permuted MNIST.

| Task 1 | Task 2 | Forgetting (F1) | $\lambda_1^{\mathbf{max}}$ | $\|\mathbf{\Delta w}\|$ |
|---|---|---|---|---|
| Stable | Stable | $1.61 \pm (0.48)$ | $2.19 \pm (0.12)$ | $73.1 \pm (1.23)$ |
| Stable | Plastic | $4.77 \pm (1.72)$ | $2.21 \pm (0.18)$ | $174.4 \pm (7.66)$ |
| Plastic | Stable | $12.45 \pm (1.58)$ | $7.72 \pm (0.19)$ | $74.7 \pm (2.63)$ |
| Plastic | Plastic | $19.37 \pm (1.79)$ | $7.73 \pm (0.22)$ | $178.4 \pm (8.58)$ |

## C.3  Additional result for experiment 1: comparing norms of weights

In section 4.2, we discussed that dropout pushes down the norm of the weights by regularizing the activations. To support this argument, in Figure 7 we compare the norm of the weights for stable and plastic networks in experiment 1. For the stable network the norm of the optimal solutions after training each task is smaller and this might be the reason for smaller displacement in the sequential

optimizations for tasks. Further analysis and theoretical justification of this phenomena is beyond the scope of the current paper and is left as an interesting future work.

(a) Permuted MNIST

(b) Rotation MNIST

Figure 7: Comparing the norm of parameters for networks in experiment 1

## C.4 Additional result for experiment 2: comparison of the first task accuracy

In Section 5.3, we provided the plots for the evolution of average accuracy during the continual learning experience. Here, we measure the validation accuracy of the first task during the learning experience with 20 tasks in Figure 8. The figure reveals that the stable network remembers the first task much better than other methods.

(a) Permuted MNIST

(b) Rotated MNIST

(c) Split CIFAR-100

Figure 8: Evolution of the first task accuracy

## C.5 Additional result for experiment 2: CIFAR-100 with more epochs

In this experiment, we extend the number of epochs for the CIFAR-100 benchmark. Table 4 shows that the accuracy of stable-SGD increases as the number of epochs per task increases. However, we note that the single epoch setting would be more realistic as it is closer to the online, real-world setups.

Table 4: Comparison of stable-SGD performance on CIFAR-100 benchmark with different number of epochs per task

|  | Average Accuracy (%) | Forgetting |
|---|---|---|
| 1 epochs | 59.9 ($\pm$ 1.81) | 0.08 ($\pm$ 0.01) |
| 5 epochs | 64.9 ($\pm$ 0.87) | 0.13 ($\pm$ 0.01) |
| 10 epochs | 69.7 ($\pm$ 0.74) | 0.12 ($\pm$ 0.01) |

## C.6 Measuring Forward Transfer

Although our main focus in this work is on the catastrophic forgetting and the "backward transfer", we include the forward-transfer on rotation MNIST with 5 tasks. We see the forgetting for stable net is much less by compromising a negligible amount of forward transfer. This is in line with our discussion on stability-plasticity dilemma.

| (a) Stable Network | (b) Plastic Network |

Figure 9: Comparison of the forward transfer in stable and plastic networks on rotation MNIST with 5 tasks.

## C.7 New experiment: comparison with orthogonal gradient descent (OGD)

The idea behind orthogonal gradient descent (OGD) [14] is to change the gradient updates to be perpendicular to the space spanned by the gradient vectors of previous tasks with the assumption that these orthogonal updates will change the output of the network minimally. Bennani and Sugiyama [5] shows that the OGD updates can only be effective across a single task and propose OGD+ that it is robust to catastrophic forgetting across an arbitrary number of tasks with tighter generalisation bounds.

However, in practice, we find that OGD is not a strong competitor to stable-SGD. In Table 5 and Table 6, we compare the accuracy of each tasks once the network trained continually on five tasks of rotated-MNIST and permuted-MNIST, respectively. We directly report the numbers from the OGD paper as our setting was exactly the same. Moreover, we note that the reported numbers for stable-SGD are calculated over five runs, each with different seed.

| Method | Task 1 | Task 2 | Task 3 | Task 4 | Task 5 |
|---|---|---|---|---|---|
| OGD | 75.6 | 86.6 | 91.7 | 94.3 | 93.4 |
| Stable-SGD | 91.5 | 92.1 | 95.02 | 94.2 | 90.9 |

Table 5: Comparison of OGD and stable-SGD on rotated MNIST with 5 tasks

| Method | Task 1 | Task 2 | Task 3 | Task 4 | Task 5 |
|---|---|---|---|---|---|
| OGD | 79.5 | 88.9 | 89.6 | 91.8 | 92.4 |
| Stable-SGD | 94.2 | 92.1 | 92.7 | 92.3 | 91.4 |

Table 6: Comparison of OGD and stable-SGD on Permuted MNIST with 5 tasks

## C.8 New experiment: stabilizing other methods

One important question that deserves further investigation is *"Can other methods benefit from the stable training regime?"*. In this section, we show that the answer is yes.

For the rotation MNIST dataset with 20 tasks, we use similar architecture and hyper-parameters for SGD, EWC, A-GEM, and ER-Reservoir as described in Appendix B. To make these models stable, we add dropout (with the dropout probability 0.25), keep the batch size small, and use a decay factor of 0.65 to decrease the learning rate at the end of each task.

Table 7 shows that by merely stabilizing the training regime, we can improve the average accuracy of EWC, A-GEM, and ER-Reservoir by 12.9%, 15.8% and 9%, respectively. Besides, Figure 10 shows the evolution of the average accuracy throughout the learning experience. As expected, the episodic memory in stable A-GEM and stable ER-Reservoir helps these methods to outperform Stable SGD, which does not use any memory. However, as noted before, stable SGD outperforms these complex methods in their original form when they do not employ stabilization techniques.

Table 7: Stabilizing other methods: the average accuracy and forgetting on rotated MNIST with 20 tasks.

| Method | Average Accuracy | Forgetting |
|---|---|---|
| SGD | 46.3 ($\pm$ 1.37) | 0.52 ($\pm$ 0.01) |
| Stable SGD | 70.8 ($\pm$ 0.78) | 0.10 ($\pm$ 0.02) |
| EWC | 48.5 ($\pm$ 1.24) | 0.48 ($\pm$ 0.01) |
| Stable EWC | 61.4 ($\pm$ 1.15) | 0.30 ($\pm$ 0.01) |
| AGEM | 55.3 ($\pm$ 1.47) | 0.42 ($\pm$ 0.01) |
| Stable AGEM | 71.1 ($\pm$ 1.06) | 0.13 ($\pm$ 0.01) |
| ER-Reservoir | 69.2 ($\pm$ 1.10) | 0.21 ($\pm$ 0.01) |
| Stable ER-Reservoir | 78.2 ($\pm$ 0.74) | 0.09 ($\pm$ 0.01) |

Figure 10: Evolution of average accuracy

## Footnotes

[3]For further instructions for reproducing the results, see our code repository.