[Reviews · NeurIPS 2020]

Review 1

Summary and Contributions: The paper hypothesizes that geometric properties around local minima on the loss surface of neural networks, which have previously been linked in the literature to generalization, affect catastrophic forgetting. Specifically, it proposes the product of the maximum eigenvalue of the Hessian at the minimum times the L2 norm of the difference between the weights as an (approximate) upper bound on the difference in loss on a task after training on the task itself and a consecutive one as a measure of forgetting. This bound suggests the use of techniques that reduce the spectral norm of the Hessian around the loss of the minimum that training converges to, of which there are plenty in the literature. The paper makes specific suggestions regarding batch size, initial learning rate and learning rate decay, which match choices made in previous works that tuned them as hyperparameters. Further it argues for the use of dropout. It empirically demonstrates that a neural network trained using only these hyperaparameter settings can perform well on continual learning tasks, even without the use of dedicated methods for preventing forgetting. ************** POST REBUTTAL UPDATE: I remain somewhat concerned about the performance of some baselines. E.g. since EWC is just SGD with an additional L2-like regularizer from the 2nd task on, I find it hard to imagine that Stable-SGD would outperform it. Better performing baselines would not change the conclusions from the paper (how to set hyperparameters for continual learning problems and that SGD can perform surprisingly well if tuned properly), so I'm still happy to argue in favor of acceptance, but not willing to give a higher score.

Strengths: The paper is overall well-argued and -motivated and clearly written. It does a good job of synthesizing a broad range of results from the literature. While previous works on continual learning (at least the ones that I'm more familiar with) had tuned hyperparameters like learning rate, batch size etc relatively "blindly", this paper makes reasonable suggestions for how to choose them (or at least their range) in a more informed manner.

Weaknesses: The technical novelty of the paper is fairly low, but given that it builds on a rather broad selection of results from the literature and synthesizes them well, I don't find this too problematic. The presentation of the experiments is somewhat unsatisfying to me. All the techniques that the paper suggests are options that would be perfectly reasonable to consider for a continual learning method. So reporting that the "vanilla" neural network with the right hyperparameters outperforms continual learning methods leaves me with the impression that the baselines were not tuned properly. Indeed the grid search over hyperparameters for the baselines as per the appendix is not exactly exhaustive (and it would be better to do a random search over the range anyway). This has been mitigated to a degree by the additional results using the insights from the paper to improve the baselines. My suggestion would be to emphasize more clearly that the paper makes two distinct contributions: (1) plain neural networks can be much stronger baselines for continual learning than previously thought if using the right hyperparameters and (2) those hyperparameters choices are also useful improving continual learning techniques. I personally would expect the latter point to be much more relevant for the community, however this is unfortunately the point that does not come across as clearly from the way the paper is currently written. All things considered, I would see this as a borderline paper tending towards an accept. The observations are interesting and both write-up and experimental design are polished enough, although I think that the paper could be strengthened by restructuring the emphasis in the presentation of its contributions as discussed above, so I would not mind seeing an improved version at a future conference either.

Correctness: Both technical discussion and empirical methodology appear to be correct to me.

Clarity: The overall structure of the paper makes sense and the write-up is easy to follow.

Relation to Prior Work: The paper mostly synthesizes prior work and makes extensive references to the literature. A further perspective that may be interesting to discuss could be that through the Bayesian lense of doing online learning using the Laplace approximation (see also the note by Huszar on EWC [1] and the original work by MacKay[2] of course in addition to the referenced Ritter et al. paper). Derived similarly to the upper bound on forgetting in the paper through a second order Taylor expansion, the training techniques suggested in the paper would (assuming equal data fit) lead to an approximate posterior with higher model evidence. This would also make combining these hyperparameter choices with methods that approximate the curvature, such as EWC, particularly relevant. [1] F. Huszar. On Quadratic Penalties in Elastic Weight Consolidation. arXiv: 1712.03847 [2] D. MacKay. A Practical Bayesian Framework for Backpropagation Networks

Reproducibility: Yes

Additional Feedback: misc/typos: - line 144: "tight" -> "tied" - line 231: sentence is a bit clunky ("well-established" and "well-studied")


Review 2

Summary and Contributions: The authors examine a novel quantity for continual learning which they call forgetting---a loss distance measure---which they use to motivate an examination of the geometry of minima to avoid catastrophic forgetting. They then show that a number of well-studied methods for encouraging wide minima also avoid catastrophic forgetting.

Strengths: I really like this paper. I have to admit that on reading the title and abstract I expected little more than a hyperparameter search but quickly realized what a substantive contribution this paper represents! The authors' observations line up with my own experiences on effective hyperparameters for 'naive' CL baselines, with a rigorous empirical grounding and a plausible theoretical justification. As the authors note, these observations can be used to inform and improve a wide range of other approaches as well as setting a higher bar for 'naive' baselines for other methods to compare to. I would hope that all papers will soon compare to SGD baselines inspired by the results of this paper.

Weaknesses: -Later in the paper, and in other work e.g., Chaudhry [5] which you cite, "forgetting" is a measure that looks at relative accuracy (I may be misremembering, but I think their definition is slightly different again from your definition in 5.1). But in eq (2) you define something related to loss. It would probably be better to pick a new word for this. -I think you could say a little more to motivate why your new forgetting measure is an important quantity for CL. I think it is quite important, but it would be good to explain more. Otherwise you are arguing that wide minima are good for reducing F_1, but not arguing why we care about reducing F_1. (If it weren't for the fact that your provide empirical evidence suggesting this leads to improvement of more standard metrics for CL performance, I would judge this more harshly, but it should still be addressed.) -Your analysis in l163 seems to point towards early stopping as an important tool for CL, perhaps that could get more attention.

Correctness: I was generally quite impressed by the empirical evaluation. Definitely among the best of CL papers I have read.

Clarity: I think this was the weakest point of the paper actually. The abstract and intro are wordy. A number of passages read as though the authors cut individual words in order to fit the space requirements rather than cutting sentences/ideas resulting in overly terse language. E.g., the paragraph starting line 149. I was worried on first reading that the point which was made in l147 was not going to be addressed, it might be worth signposting more clearly that you intend to consider it.

Relation to Prior Work: I think this is fair rather than excellent. You probably could discuss more the other methods which implicitly rely on a similar observation. E.g., some of the regularization-based approaches might implicitly be aiming at a similar thing. But this is plausibly future work.

Reproducibility: Yes

Additional Feedback: -Table 1 appears a bit early. -l120/121 minima is used as singular, should be minimum. -As you define it the L_k in (1) is the population risk not empirical risk, I think. Perhaps be clearer about notation, seems that T is a set, but you talk about a distribution which is not defined. I know exactly what you mean by the equation, but may as well be careful. Maybe extra comma? ------------ Thank you for your response. Having considered it as well as the comments of the other reviewers, I'm slightly reducing my score. I'm slightly concerned about the fact that the results in Table 2 for some of the baselines seem noticably worse than previously published results for that same baseline. I think the paper's main point is intact whether or not your well-tuned SGD beats baselines, but you may want to either check the baselines or remove claims that SGD beats them (which would, in my view, still leave an interesting result).


Review 3

Summary and Contributions: The authors provide a theoretical study on how some known training techniques affect training in continual learning. First, they set up a mathematical framework to explain the importance of the geometry of the reached minima. Second, they explain how learning rate, batch size, optimizer, dropout and weight decay fit in this framework. Finally, they present a series of experiments, comparing a simple SGD baseline (with carefully tuned training regime) with other SOTA methods.

Strengths: The paper is well written and revolves around clear and intuitive ideas, consistently motivated by theoretical insights. The mathematical explanations that are provided in section 3 are especially interesting and easy to follow. It is interesting to see a paper that explicitly addresses how the aspects of training regimes affect continual learning. Evaluations in this area of research typically vary meaningfully across different works, so it is very important to have work that stops and reconsiders how training should be done.

Weaknesses: The experimental setting does not make clear whether task-labels are provided at test time. This is especially unclear w.r.t. To the evaluation on CIFAR-100. By reading the provided source it seems that task labels are given, but this should be clearly specified in the paper. I found the choice of only performing 1 epoch on Split-CIFAR100 questionable. I agree with the authors claiming that this is a challenging dataset and, for this reason, I would like to see the results when more epochs on it are performed. Indeed, the authors of [5] increase the number of epochs to 30 for a fair comparison with EWC. In general, it is common to perform even more epochs-per-task on this dataset: [1] performs 70, [2] and [3] perform 250 and [4] performs 160. The results of Table 2 could not be correctly reproduced by using the provided source code. By running replicate_experiment_2.sh I obtained the following accuracies for Stable SGD: 27.87 for perm-mnist; 35.95 for rot-mnist; 49.61 for cifar100 (seemingly worse than the SGD baseline on MNIST-based datasets). This is in contrast with Table 2 showing SGD outperforming all other methods and impacts majorly on section 5.3. If this is due to a bug in the provided code, it is paramount to fix it. Also, by running replicate_experiment_1.sh, I obtained 87.00 for rot-mnist (stable) and 60.65 for perm-mnist (stable) as values of mean accuracy. Looking at Fig. 3, it seems to me that these numbers should be over 90. [ 1 ] Rebuffi et al., iCaRL: Incremental Classifier and Representation Learning [ 2 ] Wu et al., Large Scale Incremental Learning [ 3 ] Abati et al., Conditional Channel Gated Networks for Task-Aware Continual Learning [ 4 ] Hou et al., Learning a Unified Classifier Incrementally via Rebalancing [ 5 ] Chaudhry et al., Efficient Lifelong Learning with A-GEM

Correctness: Claims and method seem correct. The experimental setting is also overall correct, but experiments cannot be reproduced probably due to a bug in the released code.

Clarity: The paper is very clear and well-written.

Relation to Prior Work: Yes, this work is an analysis of how network training regime affects CL. No major work tackled this problem since [20 in the paper] and the differences w.r.t. it are clear.

Reproducibility: Yes

Additional Feedback: After the rebuttal I thank the authors for the clarification. I was able for reproducing paper results for the two exps they gave me the args thus I am supposing results are coherent and sure authors will fix the code assuring reproducibility but still need some severe hyperparam and args tuning thus this strong dependecy must be accounted in the evaluation. As for the idea it stays interesting from a technical point of view. The study is indeed a contribution in better understanding training regimes for CL which is definitely an understudied and complex problem. For this reason I raised my scores of 1 tick.


Review 4

Summary and Contributions: [The problem they focus on] The paper works on alleviating catastrophic forgetting problem in continual learning. They aim to study the impact that different training techniques (dropout, learning rate decay, and batch size) on forgetting. They model the forgetting to understand the dynamics during the training and utilize training techniques to reduce the forgetting among tasks. [Basic assumption in the paper] They utilized a second-order Taylor expansion to approximate the forgetting between two tasks. They assumed that the models are overparameterized where the loss is convex in a reasonable neighborhood around their local minimum. [Their solutions] They modeled and bounded the catastrophic forgetting through an optimization and geometrical view. The proposed wide/narrow minima hypothesis is consistent with the generation theory in neural networks. Based on the theoretical bound and hypothesis, they proposed stable SGD via controlling optimization (learning rates, batch size, and optimizer) and regularization (dropout and weight decay) settings to reduce the modelled forgetting in CL domains. [The results] They perform the experiments on three standard continual learning benchmarks: Permuted MNIST, Rotated MNIST, and Split CIFAR-100. They use a feed-forward neural network with two hidden layers and a ResNet18 as their models. The evaluation metrics are average accuracy and average forgetting. The experiment results show that (1) wide/narrow minima hypothesis exists in low-dimensional continual learning tasks; (2) From table 2 and fig 4, the stable SGD is consistently outperforming some classical CL methods (EWC, A-GEM, etc) in Permuted/Rotated MNIST, Split CIFAR-100. [The contributions] They analyzed catastrophic forgetting with the loss landscape and optimization perspectives. It is shown that these simple training techniques can also improve the performances of continual learning tasks. The study provides practical insights to improve stability via simple yet effective techniques that outperform alternative baselines.

Strengths: [Theoretical guarantees] The work formalized and approximated the catastrophic forgetting using second-order Taylor expansion. They found out that the bound for forgetting between two tasks are bounded by the width of a local minima and the displacement. The derived geometrical properties (wide/narrow minima hypothesis) are consistent with the wide/narrow minima hypothesis in generation theory in neural networks. [Novelty and contribution] As far as I know, no prior work has provided comprehensive study on how training techniques (e.g., batch size, learning weight decay, dropout) affect catastrophic forgetting. The idea is novel and insightful to understand forgetting and learning dynamics in continual learning tasks. Although the work is quite empirical, the authors provided a set of widely used techniques to control stability and forgetting based on the wide/narrow minima hypothesis. They extended the prior works on the neural network generation via optimization and regularization settings into a continual-learning manner. Both some of the theoretical analysis and practical strategy are insightful for understanding the training dynamics in continual learning tasks. [Effectiveness of the proposed method] The empirical results are consistent with the proposed wide/narrow minima hypothesis under low-dimensional experiments. The proposed stable-SGD can reduce catastrophic forgetting effectively and outperform some classical CL methods under standard experimental settings. The approach is simple but efficient. The experiments are easy to follow and reproduce with the detailed information given in the supplementary materials.

Weaknesses: Indeed the proposed method works well on the continual learning benchmarks. I have following concerns and questions: [The lack of other CL metrics] I admit that this work focuses on forgetting problem in training deep neural networks via SGD manner on sequential tasks, where they focused on improving the backward transfer (the ability to capture new tasks) while ignoring to improve forward transfer or adaptation on new tasks, which is also an important metric for evaluation in CL. Many existing works (e.g., Davidson et al., CVPR 2020; Riemer et al., ICLR 2019; Díaz-Rodríguez et al., NIPS Workshop 2018) analyzed both backward forgetting and forward transfer at the same time. Analyzing training regimes using similar techniques (minima hypothesis, geometrical properties) for forward transfer among tasks is also meaningful. [Hard to extend the work with high-dimensional input] Indeed, the wide/narrow minima hypothesis is effective for low-dimensional input. However, it is hard to extend into high-dimensional problems since the bound will be loose in Equation (5). Thus It is intractable to follow the hypothesis under high-dimensional settings, which is more practical in real-world cases. [Benchmarks] I agree that most of the continual learning algorithms focus on naive benchmarks, e.g., split MNIST, CIFAR, which naturally is the classification problem. However, since the CL field has developed, we are more willing to see some practical applications for applying lots of CL strategies into some computer vision, natural language processing tasks. This work only has results on classification tasks, thus I am not sure whether it is still feasible on other practical applications, e.g., object recognition/detection, action recognition, scene segmentation. Since high-dimensional or more challenging tasks need a sophisticated design of the algorithms, thus I am not confident in the importance of this idea. Why not try more challenging benchmarks ? or integrate your idea into it, e.g., She et al., ICRA 2020 ? [Lack more rigorous theoretical analysis] Can the author provide convergence analysis on the proposed stable-SGD in continual learning, like the convergence analysis of SGD and OGD in continual learning settings proposed by (e.g., Bennani et al,, 2020)? [Lack computational metrics] Going through the paper and the provided codes, the framework requires us to first compute the spectrum of the Hessian for each task. It is better to compare the complexity (computational metrics) with other CL methods for comparison. For understanding the neural network’s optimization property, understanding the Hessian spectrum is quite important. Some prior work uses NTK (Jacot A et al,, 2020) to study asymptotic spectrum of Hessian both at initialization and during training. [Lack practical suggestions on parameter choosing] The stable SGD needs to decide optimal (or suboptimal) training techniques (learning rates, batch size, optimizer, dropout and weight decay) for the tasks with the knowledge of the spectrum of the Hessian. The parameters (optimization and regularization settings) choices vary a lot among different tasks and datasets. Thus it might not be applicable and practical for continual learning applications. Can the authors provide practical and empirical parameter choices for readers to follow the work? I also feel the work depends on the hessian spectrum while not having sufficient theoretical analysis. [Lack the analysis on other CL methods using the proposed hypothesis] Can the authors provide some analysis on the relationship between minima hypothesis and previous regularization-based CL methods (SI, EWC, etc). Can these approaches also achieve wider minima? [Lack comparisons with similar approaches] There are also some similar approaches using alternated SGD to alleviate catastrophic forgetting: e.g., OGD proposed by (Farajtabar et al., AISTATS 2020). Can the authors discuss similar approaches in the introduction part briefly and why not compare with other strategies in the experiments? Davidson, G. and Mozer, M.C., 2020. Sequential mastery of multiple visual tasks: 1. Networks naturally learn to learn and forget to forget. In Proceedings of the IEEE/CVF Conference on Computer Vision and Pattern Recognition (pp. 9282-9293). 2. Riemer, M., Cases, I., Ajemian, R., Liu, M., Rish, I., Tu, Y. and Tesauro, G., 2018. Learning to learn without forgetting by maximizing transfer and minimizing interference. ICLR 2019. 3. Díaz-Rodríguez, N., Lomonaco, V., Filliat, D. and Maltoni, D., 2018. Don't forget, there is more than forgetting: new metrics for Continual Learning. NIPS Continual Learning Workshop 2018. 4. Rao, D., Visin, F., Rusu, A., Pascanu, R., Teh, Y.W. and Hadsell, R., 2019. Continual unsupervised representation learning. In Advances in Neural Information Processing Systems (pp. 7647-7657). 5. Shui, C., Abbasi, M., Robitaille, L.É., Wang, B. and Gagné, C., 2019. A principled approach for learning task similarity in multitask learning. IJCAI 2019. 6. Farajtabar, M., Azizan, N., Mott, A. and Li, A., 2020, June. Orthogonal gradient descent for continual learning. In International Conference on Artificial Intelligence and Statistics (pp. 3762-3773). 7. Bennani, M.A. and Sugiyama, M., 2020. Generalisation Guarantees for Continual Learning with Orthogonal Gradient Descent. arXiv preprint arXiv:2006.11942. 8. She Q, Feng F, Hao X, et al. OpenLORIS-Object: A robotic vision dataset and benchmark for lifelong deep learning. ICRA 2020 9. Jacot A, Gabriel F, Hongler C. The asymptotic spectrum of the Hessian of DNN throughout training, ICLR 2020

Correctness: Yes, the claims and the empirical methodology are correct.

Clarity: Yes. The whole paper is well-written. there are some minor typos in the paper, e.g., The footnote in page 2, below line 90, Potential exceptions being the early work of Goodefellow et. al. Goodefellow -> Goodfellow; Line 98, data point -> data points Line 113, [29] -> (Jacot et al., 2018) [29]. Line 155, make the observation -> oberve

Relation to Prior Work: Yes, the paper discussed the novelty and differences from prior works clearly.

Reproducibility: Yes

Additional Feedback: The work is insightful for understanding the dynamics of training in continual learning tasks. However, most of the experiments of this paper focus on low-resolution data with supervised image classification tasks, which are a bit simple. It is meaningful to extend the work into larger real-world datasets like CORe50 and more learning paradigms (RL settings(Riemer et al., ICLR 2019), unsupervised settings(Rao et al., NeurIPS 2019)). _________________________________ update review: I have gone through the response letter from the authors, and some key concerns have been well addressed. The concerns for extending to high-dimensional input data, reporting other CL metrics, comparision with OGD, and computational cost of stable-SGD are further explained and added with more experiments by authors. Thus, raise my score to weak accept ~

[Author Response · NeurIPS 2020]

We thank the reviewers for their thoughtful and positive feedback. We are encouraged they found our work well-argued
**(R1)** and with substantive contributions **(R2)** . We are pleased **(R3)** finds our work consistently motivated by theoretical
insights and **(R4)** identifies the idea as novel and insightful. We address reviewers comments below and we make sure
to incorporate all feedbacks.

@**(R1)** - **Baselines not tuned properly**: We used the grid for hyper-params set by their authors. For example, for
A-GEM and ER-Reservoir, we had the best hyper-params reported in the appendix of the original papers in our grid.

@**(R1)** - **Emphasize on two distinct contributions**: We agree and we'll update the introduction and conclusion section.
@**(R2)** - **Motivate your new forgetting measure**: We also believe this definition needs more context. Regarding the
forgetting measure in experiments we agree it's not new and we will provide additional context to reduce the confusion.

@**(R2)** - **Early stopping as an important tool for CL**: Yes. Early stopping would prevent going further from the
previous minima. However, there will be a trade-off since if the parameters do not change enough, the new tasks can not
be learned. This is very related to the discussion in l204 as they both refer to the distance from the previous minima.

@**(R3)** - **Experimental setting**: Task labels are provided for CIFAR-100, only to have a similar implementation with
other baselines (e.g., A-GEM, ER-Reservoir). Nevertheless, we agree that this deserves explicit discussion and we
will update the paper. Regarding the number of epochs for CIFAR experiment, we did so to be compatible with all the
benchmarks as explained in the paper. However, we will add the results for more training epochs.

@**(R3)** - **Reproducibility**: We apologize if you had difficulty to reproduce the results. We uploaded the polished and
cleaned version of the code since we wanted to make sure the reviewers can investigate the code. However, there were
some minor typos in the uploaded code and we sincerely apologize for that. We wish we could upload a new version of
the code or share an anonymous link to our experiments to make it easy to verify the results. Unfortunately, according to
the NeurIPS guidelines, we can not share any external links. However, as a proof of concept, you can verify experiment
1's rotation MNIST benchmark using the following command which should give an average accuracy of 92.4% with all
accuracy metrics above 90%.
`python -m stable_sgd.main --dataset rot-mnist --tasks 5 --epochs-per-task 5 --lr 0.15 --hiddens 100 --batch-size 16 --gamma 0.25  --dropout 0.25 --seed 1234`
In addition, you can use the the following command to replicate experiment 2 on CIFAR-100:
`python -m stable_sgd.main --dataset cifar100 --tasks 20 --epochs-per-task 1 --lr 0.2  --gamma 0.85 --batch-size 10 --dropout 0.1 --seed 2345`
To ensure all results are reproducible we'll add external links to each experiment in the revision.

@**(R4)** - **Extension with high-dimensional input**: We acknowledge that our results may be affected by the curse
of dimensionality. However, we discussed this issue in lines 106-116 and especially in 129-133. Moreover, recent
work by Fort & Ganguli ("Emergent properties of the local geometry of neural loss landscapes") suggests that: (1)
The Hessian eigenspectrum is composed of a bulk plus C outlier eigenvalues where C is the number of classes, (2)
Gradient aligns with this tiny Hessian subspace which implies that most of the descent directions lie along extremely
low dimensional subspaces of high local positive curvature. Hence, the bound would be still tight enough for our sake
as also demonstrated empirically in Fig2-c,d on high dimensional models.

@**(R4)** - **Other CL metrics**: As noted by the reviewer, the focus of this work was on forgetting. However, we have
included results for forward transfer in Fig. 1 & 2 on rotation-MNIST. We see the forgetting for stable net is much less
by compromising a negligible amount of forward transfer. This is in line with our discussion on stability-plasticity
dilemma. We will include this result in the paper. Thanks for pointing out.

| Method | Average Acc |
|---|---|
| SGD | 53.9 (± 4.2) |
| Stable-SGD | 65.5 (± 3.7) |

| Method | Task 1 | Task 2 | Task 3 | Task 4 | Task 5 |
|---|---|---|---|---|---|
| OGD | 75.6 | 86.6 | 91.7 | 94.3 | 93.4 |
| Stable-SGD | 91.5 | 92.1 | 95.02 | 94.2 | 90.9 |

**Figure 1:** Plastic Net    **Figure 2:** Stable Net    **Table 1:** OGD versus stable-SGD    **Table 2:** CORe50 result

@**(R4)** - **Lack comparisons with similar approaches**: We compare our method with OGD. The validation accuracy
for each task at the end of CL experience on rotation MNIST with five tasks is reported in Table 1. We will add
discussions on OGD [Farajtabar et al] and its extensions [Bennani et al] and other papers (EWC, SI, etc) in the revision.

@**(R4)** - **Benchmarks - low-resolution data with supervised image classification tasks, which are a bit simple**:
While we agree with the reviewer that these datasets are simple, we want to highlight that they are difficult for *sequential*
*multitask* problems. For instance, the best result in our paper on rotation MNIST dataset with 20 tasks yields  78%
accuracy which leaves a lot of room for improvement. Moreover, to demonstrate the feasibility of the proposed work on
other applications we have included a preliminary result for five runs over NI setting of CORe50 dataset with Resnet18
network in Table 2. Full results will be reported in the revision.

@**(R4)** - **Computational metrics**: Computing the Hessian in the code is just for reproducing eigenvalues in Fig 3 and
empirically showing that minima found by stable-SGD is wider. It is not a part of the algorithm. Hence, stable-SGD is
as computationally cheap as normal SGD, which we believe is an advantage [over many alternatives].

[Meta-Review · NeurIPS 2020]

After much discussion, the reviewers converged towards recommending to accept this submission. The reviewers think the claims are interesting, were satisfied with the authors' response, and have updated their reviews accordingly.